# Possible Mechanisms of Oxidative Stress-Induced Skin Cellular Senescence, Inflammation, and Cancer and the Therapeutic Potential of Plant Polyphenols

**DOI:** 10.3390/ijms24043755

**Published:** 2023-02-13

**Authors:** Hui-Min Liu, Ming-Yan Cheng, Meng-Han Xun, Zhi-Wei Zhao, Yun Zhang, Wei Tang, Jun Cheng, Jia Ni, Wei Wang

**Affiliations:** 1School of Perfume & Aroma and Cosmetics, Shanghai Institute of Technology, Shanghai 201418, China; 2Engineering Research Center of Perfume & Aroma and Cosmetics, Ministry of Education, Shanghai 201418, China

**Keywords:** oxidative stress, ROS, signaling pathways, plant polyphenols, skin cellular senescence, inflammation, cancer

## Abstract

As the greatest defense organ of the body, the skin is exposed to endogenous and external stressors that produce reactive oxygen species (ROS). When the antioxidant system of the body fails to eliminate ROS, oxidative stress is initiated, which results in skin cellular senescence, inflammation, and cancer. Two main possible mechanisms underlie oxidative stress-induced skin cellular senescence, inflammation, and cancer. One mechanism is that ROS directly degrade biological macromolecules, including proteins, DNA, and lipids, that are essential for cell metabolism, survival, and genetics. Another one is that ROS mediate signaling pathways, such as MAPK, JAK/STAT, PI3K/AKT/mTOR, NF-κB, Nrf2, and SIRT1/FOXO, affecting cytokine release and enzyme expression. As natural antioxidants, plant polyphenols are safe and exhibit a therapeutic potential. We here discuss in detail the therapeutic potential of selected polyphenolic compounds and outline relevant molecular targets. Polyphenols selected here for study according to their structural classification include curcumin, catechins, resveratrol, quercetin, ellagic acid, and procyanidins. Finally, the latest delivery of plant polyphenols to the skin (taking curcumin as an example) and the current status of clinical research are summarized, providing a theoretical foundation for future clinical research and the generation of new pharmaceuticals and cosmetics.

## 1. Introduction

The skin is the largest organ in humans, accounting for approximately 10–15% of the body weight [1]. It plays a role in protection, excretion, immunity, body temperature regulation, and perception of external stimuli [2]. Environmental pollution and ozone layer destruction have recently increased the intensity of PM2.5 and ultraviolet (UV) rays in the atmosphere, thereby increasing the risk of skin cancer, dermatitis, and skin cellular senescence. This is because environmental factors stimulate the skin, leading to oxidative stress [3,4,5]. Oxidative stress is caused by ROS overproduction in biological systems, and these ROS cannot be completely removed by the antioxidant system, which is essentially a state of disturbed “redox homeostasis” [6]. On the one hand, excess ROS directly damage lipids, proteins, and DNA, causing oxidative damage to the skin. Binding of ROS to DNA leads to proto-oncogene activation, that to proteins leads to collagen degradation, and that to lipids results in lipid peroxidation and increased cell membrane permeability [6]. UV radiation causes oxidative damage to macromolecules, leading to inflammation, photoaging, and many skin cancer types [7]. On the other hand, excess ROS are involved in cell signaling pathways, altering the expression of numerous genes and leading to skin cellular senescence, inflammatory skin conditions, such as psoriasis and dermatitis, and cancers, such as melanoma and squamous cell carcinoma (SCC).

Several antioxidants are currently available to treat oxidative stress-induced skin cellular senescence, inflammation, and cancer. Biological antioxidants are defined as substances that significantly delay or prevent the oxidation of an oxidizable substrate when the concentration is lower than that of the oxidizable substrate [8]. The human body’s antioxidant system that functions against ROS mainly comprises antioxidant enzymes and non-enzymatic antioxidants capable of preventing oxidative damage in the human skin [9]. Antioxidant enzymes include superoxide dismutase (SOD), catalase (CAT), glutathione peroxidase (GSH-Px), glutathione reductase (GSH-Re), etc. [10]. Non-enzymatic antioxidants mainly are low-molecular-weight substances capable of neutralizing free radicals, including antioxidant vitamins and their derivatives, cofactors, melatonin, minerals, sulfur compounds, non-protein nitrogen compounds, and plant antioxidants [11,12,13,14,15]. Plant polyphenols are types of remarkable antioxidants in a different biological system, including red blood cells, because the ortho-phenolic hydroxyl group in the phenolic hydroxyl structure is easily oxidized into a quinone structure. This structure has considerable potential to scavenge free radicals, such as ROS [16]. Moreover, these polyphenols can activate cell signaling pathways and express antioxidant enzymes. Nonetheless, plant polyphenols are poorly soluble, easily metabolized, and have low bioavailability. Currently, several studies have investigated the oral preparation of plant polyphenols, but studies on the topical application of plant polyphenols to the skin, particularly clinical research, are insufficient. With biomaterial development, plant polyphenols could be stably and efficiently delivered to the skin, eliminating low biocompatibility barriers. This article introduces the targets of plant polyphenols in detail to prevent oxidative stress-induced skin cellular senescence, inflammation, and cancer, as well as the latest plant polyphenol delivery system. These data serve as a theoretical foundation for future clinical research, novel medicine development, and cosmetic product generation.

## 2. Sources of ROS in the Skin

### 2.1. Endogenous Factors of ROS

Endogenous ROS production involves intracellular enzymes (Figure 1), including the mitochondrial electron transport chain, xanthine oxidoreductase (XOR), NADPH oxidase, cytochrome P450 family enzymes, several peroxidases, cyclooxygenases, and lipid oxygenase [17]. These enzymes are widely disseminated in the mitochondria, endoplasmic reticulum, cell membrane, and cytosol, with mitochondria and endoplasmic reticulum being major contributors to ROS production. In the mitochondria, oxygen is converted to superoxide anion by mitochondrial complex I or III or xanthine oxidase. This anion is converted to hydrogen peroxide by SOD. Under normal conditions, hydrogen peroxide is decomposed into water and oxygen by antioxidant enzymes (TPx, GPxs, and CAT) in the mitochondria, whereas under oxidative stress, hydrogen peroxide generates ROS through the Fenton reaction [18]. In addition, during the oxidative metabolism of long-chain fatty acids, peroxisomes generate H_2_O_2_ radicals [19,20,21]. During protein production, NO• is produced while nitric oxide synthase is being synthesized [19,20,21]. Thus, endogenous ROS are mainly generated as a by-product of oxidative phosphorylation and enzymatic reactions that support aerobic metabolism.

### 2.2. Exogenous Factors of ROS

Exogenous ROS are mainly induced by environmental factors, with ultraviolet radiation being the main source. UV light is typically divided into three bands based on its biological effects: long-wave UV (UVA), medium-wave UV (UVB), and short-wave UV (UVC). UVA has a strong penetrating power and can reach the dermis, thereby tanning the skin, whereas UVB has a moderate penetrating power and can sunburn the epidermis [22]. Sunlight is the main source of ultraviolet radiation, and ultraviolet radiation reaching the earth’s surface consists of 95% UVA and 5% UVB [23]. Excessive skin exposure to sunlight generates high ROS levels and reduces GSH-Px and GSH-Re activities. ROS production by keratin-forming cells and fibroblasts is associated with UV radiation [24]. In addition, various factors, such as radiation, PM2.5, nanoparticles, chemicals, and thermal stimuli, may contribute to ROS generation. Table 1 summarizes the exogenous factors and adverse consequences of ROS production in some experiments.

Both endogenous and exogenous factors stimulate ROS generation, and thus have many adverse consequences on the skin, thereby causing skin cellular senescence, inflammation, and cancer.

## 3. Possible Mechanisms of Oxidative Stress-Induced Skin Cellular Senescence, Inflammation, and Cancer

Oxidative stress causes skin cellular senescence, inflammation, and cancer through two possible mechanisms: (1) ROS directly damage biomolecules, thereby causing oxidative damage; (2) ROS mediate cellular signaling pathways, thus affecting gene expression.

### 3.1. Oxidative Damage of Biological Macromolecules

Endogenous and exogenous stimuli produce large amounts of ROS that damage biological macromolecules, such as proteins, lipids, and DNA. This leads to skin cellular senescence, inflammation, and cancer (Figure 1).

#### 3.1.1. Biomacromolecule Damage and Skin Cellular Senescence

Oxidative stress is a crucial factor that affects skin cellular senescence. Genetic factors drive intrinsic aging, whereas the external environment drives extrinsic aging. Long-term exposure to UV radiation is the primary cause of extrinsic photoaging [17,43]. Both external and internal stimuli trigger ROS production in the skin, further reducing antioxidant levels in the skin and damaging DNA, proteins, and lipids [44]. Under this condition, the number of dermal fibroblasts and collagen and elastin synthesis in the extracellular matrix (ECM), particularly type I and type III of collagen, are reduced [45]. The skin then exhibits visible signs of aging, including thinning, wrinkles, dehydration, and loss of elasticity.

#### 3.1.2. Biomacromolecule Damage and Skin Inflammation

Exposure of the skin to endogenous or exogenous stimulation triggers excessive oxidative stress, thus inducing ROS production. ROS react with unsaturated fatty acids with acyl double bonds to generate lipid peroxidase. This enzyme forms hydrophilic pores in biological membranes, thereby resulting in increased cellular or mitochondrial membrane permeability, oxidative damage, and inflammatory reactions [46]. Furthermore, several studies have linked oxidative damage to various inflammatory skin conditions, such as psoriasis and dermatitis [47]. Urinary nitrate, malondialdehyde, and 8-hydroxydeoxyguanosine levels were higher in patients with psoriasis and atopic dermatitis (AD) than in healthy controls, which suggested that the development of inflammatory skin diseases is associated with the oxidation of nitric oxide, lipids, and DNA [48,49].

#### 3.1.3. Biomacromolecule Damage and Skin Cancer

Oxidative damage is the primary cause of melanoma and non-melanoma [e.g., SCC and BCC] [50]. Exposure to UV radiation is among the leading causes of oxidative stress-induced skin cancer. UV irradiation of the skin induces extensive ROS generation, causing nuclear DNA damage through the synthesis of pyrimidine (6–4), lobutane pyrimidine dimers (CPDS), 8-OXODG, and pyrimidinone photoproducts. When the genetic damage is not severe, the tumor suppressor gene p53 blocks the cell cycle in the G1 phase by DNA repair enzymes, repairing the mutant gene before replication. When the gene damage is severe and beyond effective repair, p53 controls *Bax* and *Bcl-2* and mediates apoptosis. However, chronic oxidative stress can activate proto-oncogenes (BRAF, N-Ras, RAC1, and PTEN) and deactivate tumor suppressor genes (p53 and PTCH) [46,51]. Thus, the mutated gene-carrying DNA enters the cell division cycle, resulting in the proliferation and metabolic cycle disorder of tumor cells.

### 3.2. Oxidative Stress-Related Signaling Pathway

ROS produced by cells exposed to oxidative stress mediate mitogen-activated protein kinase (MAPK), nuclear factor kappa B (NF-κB), phosphatidylinositol-3-kinase (PI3K)/protein kinase B (AKT)/mammalian target of rapamycin (mTOR) (PI3K/AKT/mTOR), Janus kinase/signal transducers and activators of transcription (JAK/STAT), Kelch-like epichlorohydrin-associated protein-1(Keap1)–nuclear factor(E2)-related factor 2 (Nrf2), and silent information regulator 1 (SIRT1)/forkhead box O (FOXO) (SIRT1/FOXO) signaling pathways. Thus, they regulate the expression of skin cellular senescence-, inflammation-, and cancer-related genes (Figure 2).

#### 3.2.1. Skin Cellular Senescence-Related Signaling Pathways

Like aging in other organs, skin cellular senescence is characterized by a gradual loss of function and regenerative potential, whereas cellular senescence is the result of aging when cells stop dividing and reach a state of stasis. Oxidative stress generates ROS, and increased levels of ROS lead to cellular senescence. Senescent cells accumulate in aging tissues and organs, impairing physiological processes, including regeneration, and leading to aging [52]. Compared with young cells, senescent cells exhibit increased beta-galactosidase activity and cell cycle inhibitors P21 and P16, as well as reduced lamin B1, and produce senescence-associated secreted protein (SASP) [53]. Excessive ROS promote the production of matrix metalloproteinases (MMPs) and inflammatory cytokines by activating two major SASP regulatory pathways in the senescent cells: the MAPK and NF-κB pathways [54]. Moreover, the transforming growth factor-β (TGF-β) signaling pathway, which is associated with collagen synthesis, is inhibited in the senescent cells.

More precisely, oxidative stress occurs in cells, and excessive ROS activate the MAPK signaling pathway. This pathway in turn activates the downstream transcription factor c-Jun, which forms a dimer with the transcription factor c-fos, inducing an increase in AP-1 levels. In addition, ROS can activate the NF-κB signaling pathway. AP-1 and NF-κB activation increases the expression of MMPs, which can almost completely degrade various protein components in the ECM, gradually destroying dermal integrity [54]. The long-term effects of MMPs can severely damage skin collagen and produce large amounts of abnormal elastic fibrous substances, which can cause skin cellular senescence. The expression of MMPs is increased in senescent cells, whereas that of tissue inhibitors of metalloproteinases (TIMPs) is decreased [55]. Additionally, inflammatory skin cellular senescence may be triggered when inflammatory factors intervene [56,57]. As ligands, inflammatory factors participate in cell signaling pathways and accelerate cellular senescence. The results of experiments conducted by using UVB-induced immortalized human keratinocytes (HaCaTs) and H_2_O_2_-induced human bioactive skin fibroblasts (BJ cells) as models suggested that the antioxidant AYAPE counteracts skin cellular senescence by reducing MMP-1 production in the MAPK/AP-1 signaling pathway and inhibits fibroblast aging [58]. Lee et al. found that cordycepin inhibited UVB-induced MMP expression and reduced skin cellular senescence by inhibiting the NF-κB pathway in human dermal fibroblasts (HDFs) [59].

In the human skin, TGF-β, a multifunctional cytokine controlling many aspects of cell function [60], stimulates dermal fibroblast proliferation, induces the synthesis of ECM collagen and elastin, and suppresses MMP expression [61]. The process of TGF-β signaling pathway activation is as follows. The extracellular TGF-β ligand binds to receptor type II (TβRII) on the cell membrane, which induces autophosphorylation. TβRII then activates type I receptor, which in turn phosphorylates R-Smad (Smad2/3). The phosphorylated R-Smad binds and phosphorylates co-Smad (Smad4) after activation. Finally, the Smad2/3/4 complex is transported to the nucleus to regulate transcription, upregulate P300 transcriptional activity, and promote histone H4K16 acetylation and collagen transcription. Moreover, TGF-β induces type I collagen production by stimulating the expression of the connective tissue growth factor (CTGF) gene [61,62,63]. Duncan et al., who conducted in vitro and in vivo studies using normal rat kidney fibroblasts, demonstrated that CTGF could induce collagen synthesis and that transfection with the antisense CTGF gene blocked TGF-β-mediated stimulation of collagen synthesis. In a mouse experiment, TGF-β activation of the human CTGF promoter/lacZ reporter gene was inhibited by subcutaneously injecting transgenic mice with collagen carboxy-terminal peptide (CTX), and CTGF was found to be essential for TGF-β-induced collagen synthesis in fibroblasts [64]. However, in cells exposed to UV-stimulated oxidative stress, excessive ROS and AP-1 production inhibits the TGF-β signaling pathway, reducing type I collagen synthesis and leading to skin cellular senescence [55,61].

In conclusion, skin cellular senescence is associated with the degradation of various ECM protein components. MMPs can degrade these components and are regulated by MAPK/AP-1 and NF-κB signaling pathways. The TGF-β signaling pathway can promote protein production in the ECM, but is inhibited by AP-1. In addition, inflammatory cytokine release can accelerate skin cellular senescence.

#### 3.2.2. Skin Inflammation-Related Signaling Pathways

As secondary messengers, ROS mediate pro-inflammatory signaling pathways and regulate the expression of various inflammation-related genes, thereby resulting in oxidative stress. Oxidative stress leads to oxidative damage and causes inflammatory skin diseases, such as psoriasis, solar dermatitis (SD), atopic dermatitis (AD), and contact dermatitis (CD). The development of these diseases is positively associated with the activation of MAPK, JAK/STAT, and NF-κB signaling pathways.

Psoriasis is a chronic inflammatory skin problem with a high global prevalence [65]. Although psoriasis is a multifactorial disease with its exact etiology unknown, its onset and progression are related to the activation of inflammatory signaling pathways [66,67,68]. NF-κB is a multifunctional transcription factor regulating the gene expression of various cytokines and inflammatory mediators. Thus, it regulates immune responses, participates in intracellular signal transduction, and regulates tissue cell pathology and physiology [69]. Various cytokines and adhesion molecules, including interleukin (IL)-1, IL-2, IL-6, IL-8, IL-12, IL-18, TNF-α, and nerve growth factor, are abnormally expressed in the lesions of psoriasis patients, suggesting that psoriasis is associated with the release of NF-κB-regulated inflammatory cytokines [70]. NF-κB is present in the promoters or enhancers of genes encoding these factors. In the absence of exogenous stimulation, IκB exists in the cell cytoplasm in an inactive form and binds to NF-κB, preventing NF-κB pathway activation. Under oxidative stress, ROS mediates TAK1 and RIP1 in cells, and these kinases phosphorylate the downstream NEMO/IKKα/IKKβ complex. The phosphorylated complex then further phosphorylates the downstream IκB, leading to ubiquitination degradation and release of the p50/p65 (NF-κB) complex involved in nuclear transcriptional regulation. Studies have reported that the CpG oligodeoxynucleotide induces psoriasis by expressing IL-17A through the NF-κB signaling pathway [71]. Moreover, psoriasis development has been associated with the MAPK signaling pathway. Keratinocyte hyperproliferation and neutrophil infiltration at the lesions are among the crucial pathological changes in psoriasis [72]. Studies have shown that MAPK can stimulate the expression of inflammatory factors or chemokines by regulating downstream transcription factors, thereby regulating Th cell differentiation and promoting the chemotaxis of T cells, DC cells, and neutrophils [73]. Moreover, MAPK promotes ki-67, PCNA, and keratin expression in keratinocytes, thus promoting keratinocyte proliferation. MAPK was significantly higher in psoriasis patients than in people with normal skin, and phosphorylated ERK was highly expressed in the spines of psoriatic lesions and the basal nucleus [74,75]. Other studies have found that total protein and phosphorylated STAT3 levels are significantly higher in psoriatic lesions than in the normal skin, and high expression of active STAT3 can lead to typical psoriasis-like symptoms in mice [76,77]. Furthermore, psoriasis-related factors, such as IL-17A, IL-17F, IL-22, and IL-23R, can be transcriptionally activated by STAT3 [78], suggesting that STAT3 plays a promoting role in psoriasis development. Thus, STAT3 is an important target for psoriasis treatment.

SD, AD, and CD are also associated with inflammatory signaling pathways. The acute response of the skin to UV rays is inflammation, such as erythema and edema, and ROS production. Oxidative stress-induced ROS can initiate the activation of inflammatory and pro-inflammatory cytokines [56]. Several pro-inflammatory signaling pathways, including NF-kB, MAPK, and JAK/STAT, are activated. Many proinflammatory cytokines and environmental stress can trigger the p38 MAPK cascade. Studies have suggested that the p38 MAPK pathway may be involved in UV-induced inflammation by regulating cyclooxygenase 2 (COX-2) activity [79,80,81], which causes SD. Other studies have observed NF-κB signaling pathway activation, expression of pro-inflammatory mediators, and inflammatory skin lesions in hairless mice chronically exposed to UVB radiation [82]. Genetic studies of UV-irradiated human epidermal cells have revealed that increased inflammatory processes were associated with the upregulation of the JAK/STAT signaling pathway [83]. AD, also known as atopic eczema, is a common chronic inflammatory skin disease characterized by recurrent, pleomorphic skin lesions; dry skin; and intense itching [84]. External application of an oligodeoxynucleotide ointment that induces NF-κB activation could improve dermatitis symptoms in the NC/Nga atopic mouse model, indicating that NF-κB deficiency was related to AD pathogenesis [85]. CD is an immune system-mediated inflammatory skin problem [86]. In the initial stage of allergic CD in mice, haptens activated Langerhans cells, while dendritic cell activation occurred mainly through the activation of MPAK/p38, suggesting that CD pathogenesis is related to the MAPK signaling pathway [87]. Furthermore, the production of IL-22, an immunoregulatory cytokine associated with CD development, was dependent on TLR4, MAPK/p38, NF-κB, and JAK/STAT signaling pathways [88].

In conclusion, oxidative stress-induced skin inflammation is associated with the release of cellular inflammatory factors (e.g., IL-2, TNF-α, and COX-2), and the expression of these factors is regulated by NF-κB, MAPK, and JAK/STAT signaling pathways.

#### 3.2.3. Skin Cancer-Related Signaling Pathways

ROS promotes the abnormal proliferation, metastasis, and penetration of various tumor cells by activating oxidative stress-related pathways, such as the PI3K/AKT/mTOR, MAPK, NF-κB, and JAK/STAT pathways [51]. Two main types of skin cancer are caused by oxidative stress, namely, melanoma and non-melanoma skin cancers (NMSCs).

Both melanoma skin cancers and NMSCs are common and have high metastatic rates. NMSCs mainly include BCC and SCC, accounting for 70% and 25% of NMSCs, respectively [89]. Skin cancer is associated with tumor cell proliferation, metastasis, and angiogenesis [90], and oxidative stress in the skin activates multiple cancer-related signaling pathways. Studies have linked skin cancer development to MAPK/ERK [91,92,93]. The study results revealed that most melanoma patients carry *BRAF* mutations that may activate the MAPK/ERK signaling pathway [94]. This entire pathway is mainly composed of protein kinases, such as RAS→RAF→MEK→ERK [95], and is sequentially activated by catalytic phosphorylation of superior protein kinases. When constitutively activated, the MAPK/ERK signaling pathway promotes melanoma development by increasing cell proliferation and tumor invasion and metastasis, and inhibiting apoptosis [96,97,98]. Moreover, the MAPK/ERK signaling pathway was activated in both mouse and human cutaneous basal melanomas, suggesting its involvement in BCC development [99].

PI3K is a specific kinase that catalyzes phosphatidylinositol lipids. This heterodimer has a P110 catalytic subunit and a P85 regulatory subunit, with protein and lipid kinase activities [100]. Akt, a serine/threonine protein kinase, is a crucial downstream effector molecule of PI3K. After activation through phosphorylation, PI3K can regulate the cell cycle by inactivating various apoptotic effector molecules, activate telomerase activity, and promote tumor growth and invasion [101]. PI3K/Akt activation occurs as follows. Under normal conditions, lipid phosphatase (PTEN) metabolizes (dephosphorylates) PIP3 back to the metabolite PIP2, thereby terminating the PI3K signaling pathway. Under oxidative stress, PI3K phosphorylates PIP2 to PIP3. The phosphorylated phosphogroup 3 of PIP3 can simultaneously recruit PDK1 and AKT proteins to the plasma membrane, causing PDK1 to phosphorylate threonine (T308) at position 308 of the AKT protein and leading to partial AKT activation. Activated AKT further activates downstream regulatory signaling pathways, including mTOR, MAPK, FOXO, NF-κB, and p53 pathways. Therefore, the tumor suppressor gene PTEN reduces tumor cell adhesion and viability and prevents their invasion and metastasis by regulating the PI3K/Akt pathway. In addition, mTOR is among the key regulatory molecules downstream of AKT, and activated mTOR promotes cancer cell proliferation and angiogenesis.

The JAK/STAT pathway is involved in inflammation and immunity associated with various human pathologies, including cancer [102]. The basic transduction process of this pathway is as follows. Binding of the ligand to the receptor dimerizes the receptor, and the receptor-bound JAK is activated through self-phosphorylation [103,104]. STAT then binds to the receptor and is dimerized, which is then activated by JAK phosphorylation. The phosphorylated STAT dimer is finally translocated to the nucleus, where it binds directly to DNA [105,106]. STAT family proteins act as both cytoplasmic signaling proteins and nuclear transcription factors. As nuclear transcription factors, they activate many different gene types, including some involved in tumorigenesis, such as anti-apoptotic genes (Bcl-2, Bcl-XL) [107,108], cell cycle regulatory genes (c-Myc, CycD) [109], lipid metabolism genes, and cell differentiation genes (AOX, GFAP) [110]. Genes involved in the JAK/STAT signaling pathway are highly expressed in tumors and SCCs at a high metastasis risk [111]. JAK/STAT also endorses the growth and metastasis of carcinogenesis-related skin cancer cells [111,112,113,114].

The NF-κB pathway allows the survival, proliferation, anti-apoptosis, and metastasis of melanoma cells [115]. Growing evidence has shown that NF-κB target genes can significantly promote cell survival. Moreover, ROS can activate NF-κB signaling to promote angiogenesis and melanoma progression [116]. Recent reports have also suggested that the p65 subunit of NF-κB is critical for skin cancer development in mice. p65 loss prevented SCC development and tumor progression [117].

Conclusively, skin cancer is related to tumor cell proliferation, invasion, metastasis, and angiogenesis. The MAPK/ERK signaling pathway enhances cell proliferation, invasion, and metastasis. PI3K/AKT/mTOR and NF-κB signaling pathways promote cell proliferation and angiogenesis. The JAK/STAT signaling pathway expresses the anti-apoptotic genes (Bcl-2 and Bcl-XL) and regulates the cell cycle.

#### 3.2.4. Antioxidant Defense-Related Signaling Pathway

Skin cellular senescence, inflammation, and cancer are associated with ROS generation. Efficient ROS detoxification is the only means to prevent, or at least limit, ROS-induced damage in cells and is achieved through low-molecular-weight antioxidants, the tripeptide glutathione, and ROS detoxification enzymes and antioxidant proteins [118]. Various NADPH-triggered cytoprotective proteins include quinone oxidoreductase-1 (NQO1), heme oxygenase-1 (HO-1), γ-glutamylcysteine synthase (γ-GCS), SOD, GSH-Px, and glutamate sulfur transferase (GST), which are regulated by Nrf2 and SIRT1/FOXO signaling pathways.

The Nrf2 signaling pathway is a key pathway regulating the cellular oxidative stress response. The downstream phase II metabolic enzymes and antioxidant proteins/enzymes regulated by KEAP1 play a crucial role in cellular defense and protection [119]. Nrf2, which belongs to the CNC regulatory protein family, is a transcription factor with a basic leucine zipper structure [120]. It is widely distributed in various human organs and is the main regulator of cellular redox reactions. Under normal physiological conditions, Nrf2 mainly binds to its inhibitor Keap1, which exists in an inactive state in the cytoplasm. Once oxidative stress occurs, Keap1 degrades rapidly through the ubiquitin-proteasome pathway, thus maintaining the low transcriptional activity of Nrf2 [121]. In cells stimulated by ROS or other nucleophiles, Nrf2 is uncoupled from Keap1. The activated Nrf2 enters the nucleus and binds to Maf protein to form a heterodimer. This heterodimer t binds ARE to induce target gene expression and regulate the transcriptional activity of phase II enzymes, antioxidant enzymes, or drug transporters to exert antioxidant effects [122,123]. When activated, the Keap1–Nrf2/ARE signaling pathway can initiate the expression of multiple downstream target proteins. These activated target proteins can regulate the body’s redox balance, allowing the body to recover from an oxidative stress state to a normal physiological state. The Nrf2-regulated downstream target proteins have been confirmed to be categorized into metabolic phase II enzymes, antioxidant proteins/enzymes, proteasome/molecular chaperones, anti-inflammatory factors, and metabolic phase III enzymes (drug transporters). For example, the NADPH-regulated downstream target proteins include NQO1, HO-1, γ-GCS, and so on.

The activity of FOXO, a crucial autophagy regulator, is mainly regulated through SIRT1 deacetylation. The SIRT1/FOXO pathway is activated by cellular oxidative stress and may allow the development of new strategies for the prevention and treatment of skin cellular senescence, inflammation, and cancer. This pathway protects cells in two main ways. On the one hand, SIRT1-regulated FOXO transcription factors can promote autophagy in oxidation-damaged biomacromolecules. Autophagy is a vital basic metabolic mechanism that maintains cell homeostasis through phagocytosis and degradation of unnecessary or dysfunctional cellular components in duplex autosolutes [124]. α-Neoendorphins can reduce UVB-induced skin photoaging by activating autophagy [125]. The expression of lysosomal cathepsins was decreased in the skin of patients with psoriasis and AD. Impaired autophagy may lead to inflammation and impaired keratinocyte differentiation, whereas stimulation of autophagy may have a therapeutic potential for inflammatory skin diseases [126]. Srivastava et al. discovered that the FOXO transcription factor mediated the anti-angiogenic effect of resveratrol and effectively inhibited tumor development and metastases [127]. On the other hand, FOXO can activate antioxidant enzymes, such as SOD and CAT, to eliminate oxidative stress-induced ROS generation [128,129].

In conclusion, both Nrf2 and SIRT1/FOXO signaling pathways can express antioxidant enzymes to reduce oxidative stress-induced ROS production, thereby preventing ROS-mediated oxidative damage. In addition, the SIRT1/FOXO signaling pathway protects cells through autophagy, thereby reducing skin cellular senescence, inflammation, and cancer.

## 4. Therapeutic Potential of Plant Polyphenols

### 4.1. Health-Promoting Benefits of Natural Products

In previous studies, it has been found that natural products play an important role in human health. Amin^.^ et al. observed that Chlorella has a protective effect on the pancreas of diabetic animals, which indicates that Chlorella can improve the condition of diabetic patients by regulating insulin secretion [130]. In addition, the plant extract Miswak has been shown to have antioxidant and anti-angiogenic effects, and is a potential antioxidant and antitumor agent [131]. Another study showed that dandelion can prevent oxidative stress, liver fibrosis, and inflammatory response in rats [132]. All these studies showed that natural products are helpful to human health and have great development potential.

### 4.2. Structure and Classification of Plant Polyphenols

The nutritional and medicinal value of active components of natural plants, such as polyphenols, polysaccharides, proteins, and alkaloids, are commonly known; they are beneficial for the treatment and prevention of skin oxidative damage [133]. Because of their wide variety of sources and high biological action, plant polyphenols are selected as possible phytochemicals for the treatment of oxidative stress-induced skin cellular senescence, inflammation, and cancer. Plants producing fruits, vegetables, and cereals contain polyphenols. Polyphenols are predominantly composed of at least two benzene rings and one or more hydroxyl substituents [134]. Owing to the vast distribution and diversity of plant polyphenols, they are primarily divided into six categories according to their chemical structures, namely, flavonoids, phenolic acids, stilbenes, lignans, tannins, and curcuminoids [135], with flavonoids being the most extensively distributed.

### 4.3. Antioxidative, Anti-Inflammatory, and Anticancer Activities of Plant Polyphenols

Owing to their unique structures, plant polyphenols possess antioxidant, anti-inflammatory, and anticancer properties. Phloretin, a polyphenol found in apples, exhibit anti-inflammatory and anticancer activities and can prevent cancer cell growth and migration [136]. Cactus produces numerous phenolic compounds, including quercetin, isorhamnetin, and kaempferol derivatives that assist in regulating the infiltration and production of soluble inflammatory mediators in cells and, therefore, play a major role in inflammation [137]. Coumaric acid possesses strong free radical scavenging activity and, hence, can lower the negative consequences of oxidative damage, including arthritis and cardiovascular disease [138]. Flavonoids limit lysozyme, β-glucuronidase, and arachidonic acid production, as well as modulate the expression of genes encoding cytokines and pro-inflammatory molecules [139]. Proanthocyanidins may be used for enhancing body functioning, such as lowering inflammation, alleviating oxidative stress, combating cancer, and influencing the immune system [140]. Phenolic acids, such as chlorogenic acid, positively contribute to improving the intestinal health of animals and enhancing the body’s antioxidant capacity [141].

### 4.4. Regulatory Mechanism of Plant Polyphenols

Based on the classification and structure of plant polyphenols, seven types of polyphenols were selected for discussion. Among them, flavonoids include catechin and quercetin, phenolic acids include ellagic acid, stilbenes include resveratrol, tannins include proanthocyanidins, lignans include honokiol, and curcuminoids include curcumin. Table 2 summarizes their molecular targets or action mechanisms of preventing oxidative stress-induced skin cellular senescence, inflammation, and cancer.

#### 4.4.1. Curcumin

Curcumin is a natural plant polyphenol and an orange-yellow diketone pigment found in turmeric rhizomes [220,221]. It exhibits anti-inflammatory, antitumor, and antioxidant properties [222] and can chelate iron ions [143].

Studies have investigated the role of curcumin in oxidative stress-induced skin cellular senescence. UV irradiation and hydrogen peroxide stimulation can induce oxidative stress in human immortalized epidermal keratinocytes (HaCat) and HDFs, as evidenced through elevated levels of ROS and malondialdehyde (MDA), decreased antioxidant enzyme activity, DNA damage, protein carbonylation, and apoptosis [223,224,225,226]. Curcumin not only reduces the aforementioned oxidative damage, but also activates TGF-β and Nrf2 signaling pathways, enhances collagen production and antioxidant enzyme expression, and decreases MMP-1 and MMP-3 levels [146,223,224,227].

Furthermore, curcumin can prevent and reduce oxidative stress-induced skin inflammation. Curcumin inhibited NADPH oxidase and reduced endogenous ROS, thereby inhibiting lipopolysaccharide-induced inflammation [144]. Moreover, curcumin alleviated skin damage in psoriatic mice by decreasing IL-22 and IL-18 expression with a 47% inhibition rate [151]. This study also reported that curcumin protected the skin by neutralizing free radicals and inhibiting MAPK and NF-κB signaling pathways. Thus, AP-1 transcription and production of cytokines (e.g., IL-33, TNF-α, IL-4, IL-5, IL-13, and IL-31) were inhibited by curcumin, thereby reducing skin inflammation in psoriatic and AD mice [150,151,152,228].

Moreover, various studies have demonstrated that curcumin has anticancer properties. Kakar et al. found that curcumin suppressed TPA-induced mRNA expression of proto-oncogenes (C-FOS, C-JUN, and C-MYC) and promoted apoptotic p53 protein expression in mice skin. Furthermore, curcumin inhibited the PI3K/AKT/mTOR pathway in SCC [158]. Because of its poor water solubility and low biological rate, researchers developed hyaluronan, nano-microemulsion, and hydrogel to enhance the biocompatibility and more effectively exert the antiaging, anti-inflammatory, and anticancer properties of curcumin [227,229,230,231].

#### 4.4.2. Catechins

Many types of catechins exist, with the most common being epigallocatechin gallate (EGCG) [161]. Green tea, apples, grapes, and other plants are the primary sources of catechins [232]. In 32 random volunteers who drank 3 cups of green tea per day for 2 weeks, green tea enhanced the body’s ability of free radical elimination [163]. Additionally, epicatechin inhibits UVA-mediated unstable iron release and reduces oxidative damage. In addition, EGCG protected rat skin and keratinocytes against oxidative stress [159,160,161,162].

Several studies have investigated the effect of catechins on oxidative stress-induced skin cellular senescence. According to Wagn et al., when combined with YAG laser, catechin can reduce wrinkles and pigmentation, increase collagen, and enhance SOD activity in the skin of photodamaged mice [233]. Catechins isolated from green tea and elm root bark can inhibit MMP-1 expression and may have antiaging effects on fibroblasts, HaCaT cells, and mice [164,165,166,167]. Green tea catechins were found to prevent skin cellular senescence by regulating NF-κB, AP-1, and MAPK signaling pathways [164].

Some studies have demonstrated that catechins can prevent and reduce oxidative stress-induced skin inflammation. Catechin supplementation prevented UV radiation-induced inflammatory erythema [234]. Catechin derivatives suppressed the cytokines TNF, IL-1, and IL-4 in hapten-induced contact dermatitis models [168]. Psoriasis is a multifactorial, complex disease characterized by an inflammatory skin response. Catechins reduced the inflammatory response in psoriatic mice [169,170].

Moreover, several studies have reported the anticancer role of catechins. By regulating antioxidant and inflammatory biomarkers, the catechin emulsion gel inhibited DMBA/TPA-induced cutaneous SCC in mice [172]. Wu et al. observed that EGCG suppressed *Bcl-2* expression to trigger death in melanoma cells [173].

#### 4.4.3. Resveratrol

Resveratrol, a polyphenol belonging to the stilbene family, is found in various plants, including grapes, soybeans, blueberries, plums, and peanuts [235,236,237,238]. When the skin is exposed to certain environments, such as sunlight or pollution, or under pathological conditions that induce oxidative stress, resveratrol at non-toxic doses (20–100 μM) can activate the Nrf2 signaling pathway. Among the genes downstream of Nrf2, glutamylcysteinyl ligase and glutathione peroxidase-2 were induced at mRNA and protein levels, thereby increasing cellular antioxidant levels [239].

Several studies have investigated the function of resveratrol in oxidative stress-induced skin cellular senescence. In a H_2_O_2_-induced model of human keratinocyte senescence, resveratrol activated FOXO3 and enhanced the expression of target genes, including catalase. Moreover, the senescent cells lacked SIRT1 compared with the non-senescent keratinocytes. This suggested that resveratrol prevents skin cellular senescence by activating the SIRT1/FOXO pathway [178]. Furthermore, resveratrol inhibited the MAPK signaling pathway, reduced COX-2 and MMP expression, and increased SOD, CAT, GSH-Px, and collagen I expression, and thus exerted its antiaging activity [174,175,176,177,178].

In addition, resveratrol can prevent and reduce oxidative stress-induced skin inflammation. PM (particulate matter) causes inflammation in human keratinocytes and is associated with various oxidative stress-induced skin diseases. Shin et al. discovered that resveratrol lowers inflammation by blocking PM-induced ROS, regulating MAPK activation, and decreasing the expression of proinflammatory cytokines [240]. Dinitrochlorobenzene induced AD in NC/Nga mice. In the experimental group, rice containing resveratrol was applied to the backs of the mice. The experimental group mice scratched less frequently, had less severe dermatitis, and had improved transepidermal water loss and skin hydration [180]. Another study showed that resveratrol downregulated NF-κB, thereby reducing inflammation in mice with imiquimod-induced psoriasis [181].

Moreover, studies have confirmed the anticancer properties of resveratrol. The occurrence of cancer is related to the abnormal proliferation of cancer cells. Resveratrol promotes cancer cell apoptosis by activating p53 [183]. Resveratrol inhibited Bcl-2 expression, promoted Bax expression, and induced programmed cell death [182]. Furthermore, resveratrol regulated apoptosis in human epidermoid carcinoma A431 cells through JAK/STAT and mitochondria-mediated pathways [184].

#### 4.4.4. Quercetin

The natural product quercetin (QC) is a flavonoid widely present in many fruits and vegetables (e.g., onion and apple) [241]. It is an antioxidant; however, maintaining QC in the skin in vivo is difficult owing to its high lipid solubility. On evaluating the biological activity of QC microcapsules (TFcQCMC) in a UVB-irradiated mouse model, Vale et al. found that TFcQCMC inhibited antioxidant degradation, 2,2’-azinobis free radicals, and reactive peroxidation hydrogenase and decreased iron concentration. TFcQCMC also decreased inflammatory cytokine production, MMP-9 activity, skin edema, collagen fiber damage, myeloperoxidase activity/neutrophil recruitment, and mast cell and sunburn cell numbers [242]. QC at 5 μg/mL effectively neutralized free radicals in HaCaT and THP-1 cells.

Several studies have investigated the role of QC in oxidative stress-induced skin cellular senescence. On studying the molecular mechanism underlying QC’s action against UV-induced skin cellular senescence, Shin et al. suggested that QC suppressed UV-induced AP-1 and NF-κB transcription and MMP-1 and COX-2 expression and prevented collagen degradation in human skin tissue. Additional investigation revealed that QC reduced ERK, JNK, AKT, JAK2, and STAT phosphorylation [185]. Evidence indicated that SIRT1 regulates cellular senescence and multiple aging-related cellular processes [243].

Moreover, some studies have demonstrated that QC can prevent and minimize oxidative stress-induced skin inflammation. QC suppressed poly(dA:dT)-induced IL-18 secretion in human keratinocytes by inhibiting the IFN-γ-induced JAK2/STAT1 pathway and downregulating AIM2 and pro-caspase-1 expression [191]. In psoriatic and AD mice, QC quercetin reduced the expression of cytokines, such as COX-2, TNF-α, and IL-1, downregulated inflammation-related NF-κB and MAPK pathways, and upregulated the expression of antioxidant enzymes, such as SOD, CAT, and GPx [192,193,194]. QC has been shown to reduce redness, itching, and inflammation in damaged areas of the human skin [244].

Moreover, several studies have reported the role of QC against cancer. Kim et al. discovered that quercetin enhanced levels of the pro-apoptotic protein Bax, phosphorylated-JNK, and phosphorylated-p38, while drastically decreasing the A375SM tumor volume [195].

#### 4.4.5. Ellagic Acid

Ellagic acid, a polyphenol belonging to the phenolic acid family, is produced by various plants, including pomegranate, strawberry, raspberry, cranberry, and grape [245,246,247,248]. Ellagic acid inhibited UV/hydrogen peroxide/MGO-induced photodamage, photoaging, inflammation, and apoptosis in HaCat cells [197,200,249,250,251].

Ellagic acid and dihydromyricetin synergistically acted against UVB-induced photoaging in mice, presumably by activating the TGF-β pathway [200].

In addition, the effects of ellagic acid on TNF-α/IFN-γ-stimulated HaCaT keratinocytes and dust mite-induced AD skin damage have been investigated in NC/Nga mice. By regulating critical inflammatory signaling pathways, such as MAPK and STAT signaling, and transcriptional activators, ellagic acid suppressed inflammatory responses [252].

Khwairakpam et al. discovered that pomegranate has potential anticancer properties. It downregulates multiple signaling pathways, such as NF-κB and PI3K/AKT/mTOR, and decreases the expression of downstream cancer genes, including antiapoptotic genes, VEGF, c-met, cyclins, Cdks, and proinflammatory cytokines [207].

#### 4.4.6. Honokiol

*Magnolia officinalis* is a physiologically active lignan utilized for centuries in traditional Chinese medicine. It exerts various pharmacological properties, including anticancer, anti-inflammatory, and anxiolytic effects [253].

In UV-induced mouse and keratinocyte aging models, honokiol reduced MMP expression by inhibiting MAPK nuclear translocation and activated NF-κB [216,217].

Honokiol also inhibits inflammatory factors in AD mice, and honokiol acetylation or methylation can improve skin transmission and anti-inflammatory effects [254,255,256].

Honokiol is also very effective in reducing skin cancer incidence. Studies have shown that honokiol reduced skin cancer diversity by 49–58% and tumor volume by 70–89% compared with the control. The 30 μg dose exhibited chemopreventive effects, while the 60 μg dose significantly reduced cancer incidence by 40% compared with the control [257]. Honokiol downregulated the expression of inflammatory factors and cyclin D1, cyclin D2, cdk2, cdk4, and cdk6 proteins, whereas it upregulated the expression of cdk inhibitory proteins p21 and p27, thus inducing apoptosis and DNA fragmentation [258,259,260,261].

Cumulatively, the aforementioned research results demonstrate that plant polyphenols can be used to treat oxidative stress-induced skin cellular senescence, inflammation, and cancer, primarily through the five routes outlined below (Figure 3). Route 1: Plant polyphenols reduce endogenous ROS generation, including chelation of iron ions, to inhibit the Fenton reaction on the electron transport chain and inhibit intracellular oxidase activity. Route 2: The structural peculiarities of polyphenols allow them to directly neutralize excess ROS. Route 3: Polyphenols can directly reduce the oxidative damage caused to macromolecular substances. Route 4: Polyphenols can block ROS-mediated signaling pathways associated with skin cellular senescence, inflammation, and cancer. Route 5: Polyphenols can activate the cellular antioxidant system signaling pathway to produce antioxidant enzymes.

#### 4.4.7. Proanthocyanidins

Proanthocyanidins, also known as condensed tannins, are oligomers or polymers of the flavan-3-ol moiety. They are commonly found in fruits, vegetables, grains, nuts, and leaves [262,263]. Proanthocyanidins may be used for enhancing body functions, such as anti-inflammation and anticancer activities, decreasing oxidative stress, and regulating immunity [140].

Proanthocyanidin is a crucial player in fighting skin cellular senescence. Oligomeric proanthocyanidin nanoliposomes reduced UV radiation-induced damage to epidermal epithelial cells (HFF-1) and had strong protective effects on collagen degradation and/or synthesis following UVA exposure. In addition, proanthocyanidin cream reduced UV-induced UPE and protected the skin from UV ray-induced harm [264].

By inhibiting the release of inflammatory substances through the MAPK and NF-κB signaling pathways, proanthocyanidins can alleviate inflammatory skin damage, such as dermatitis and psoriasis [209,210,211].

Proanthocyanidins may inhibit MAPK/ERK to limit skin cancer cell proliferation, reduce the release of NF-κB pathway inflammatory factors, and increase cancer cell autophagy [212,213].

### 4.5. Delivery Systems for Topical Use of Plant Polyphenols

Although plant polyphenols have considerable bioactivity, their bioavailability is minimal, as most polyphenols are poorly soluble and unstable [265]. At present, cutaneous delivery has emerged as among the most attractive routes in the cosmetic and pharmaceutical industries. This delivery route can overcome some disadvantages of oral drug delivery, such as low bioavailability, metabolic interactions, and cytotoxicity, while ensuring that the polyphenol is continuously and stably released at the site of action. Nevertheless, normal skin possesses a formidable barrier to drug absorption, mostly because of its particular lipid composition and stratum corneum tissue. To improve polyphenol solubility and bioavailability and to offer site-specific drug delivery with enhanced pharmacokinetic features, nanoengineered polyphenol delivery systems must be created urgently. Gels, particularly hydrogels, are the most prevalent type of topical application to date. However, development of additional delivery strategies, including lipid and polymer nanoparticles, microparticles, and transsomes, is underway (Table 3).

### 4.6. Clinical Evidence That Plant Polyphenols Prevent Oxidative Stress-Induced Skin Cellular Senescence, Inflammation, and Cancer

At present, preclinical research—that is, research on skin cells and animal skin tis-sues—has revealed that plant polyphenols prevent skin cellular senescence, inflammation, and cancer caused by oxidative stress. However, human clinical trials are relatively insufficient. Table 4 summarizes some human clinical trials. The findings of these studies revealed the following: (1) Plant polyphenols are mostly used as topical or ingested products in clinical research. (2) To ensure the therapeutic effect, plant polyphenols are tested in clinical trials at high concentrations and for longer periods, which may cause irritation to the skin, intestines, and stomach. (3) In the clinic, plant polyphenols are often combined with other compounds to improve their bioavailability. (4) More diversified and objective evaluation methods of clinical results should be established.

## 5. Conclusions

To summarize, oxidative stress is extremely harmful to the skin and is associated with the onset and progression of skin cellular senescence, inflammation, and cancer. Plant polyphenols can prevent this skin damage through their molecular targets and have attracted great research owing to their low toxicity. The most recently discovered skin drug delivery technology for plant polyphenols is more durable and effective. This system will surely encourage clinical research on plant polyphenol-based products. In the near future, the limitations associated with the use of plant polyphenols in skin applications will be overcome, and new skin products will be brought to the market and society.

## Figures and Tables

**Figure 1 ijms-24-03755-f001:**
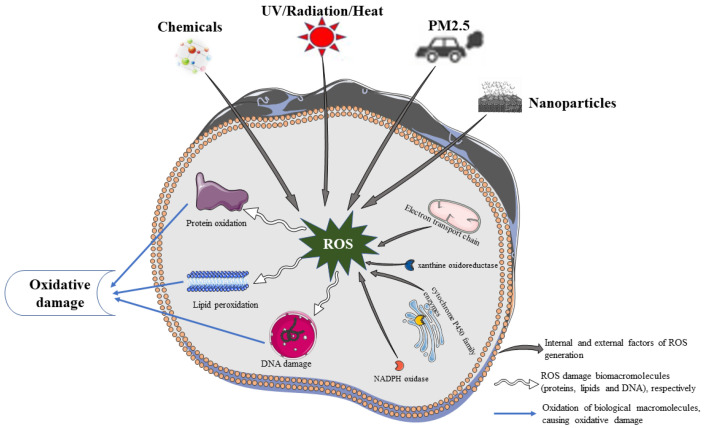
Oxidative damage of biological macromolecules.

**Figure 2 ijms-24-03755-f002:**
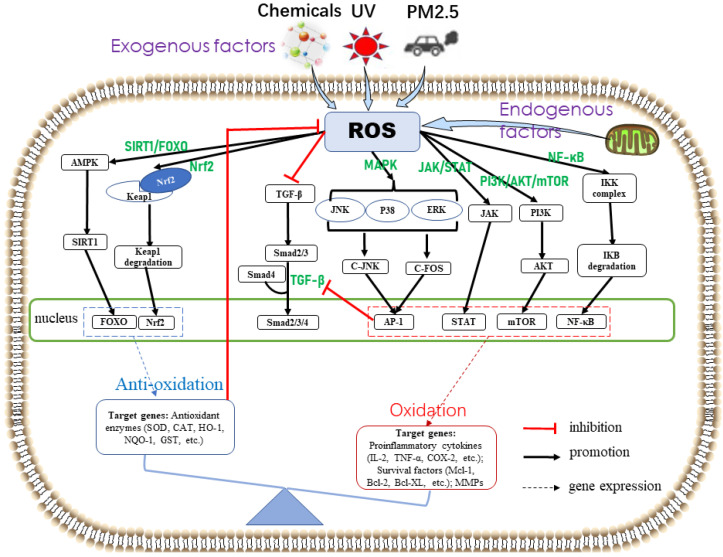
Skin oxidative stress-related signaling pathways. Exogenous stimuli (e.g., UV, chemicals, and PM2.5) and/or endogenous factors (e.g., mitochondrial oxidase and metabolism) induce ROS overproduction. ROS activate the antioxidant defense system and promote the expression of antioxidant enzymes through Nrf2 and SIRT1/FOXO pathways, which in turn reduce ROS production. When the antioxidant defense system fails to completely scavenge ROS, oxidative stress is induced. Excessive ROS activate MAPK, JAK/STAT, PI3K/AKT/mTOR, and NF-κB signaling pathways, thereby resulting in the expression of cytokines associated with skin cellular senescence, inflammation, and cancer. The transforming growth factor-β signaling pathway, related to collagen synthesis, is inhibited by ROS and the transcription factor AP-1 to accelerate skin cellular senescence.

**Figure 3 ijms-24-03755-f003:**
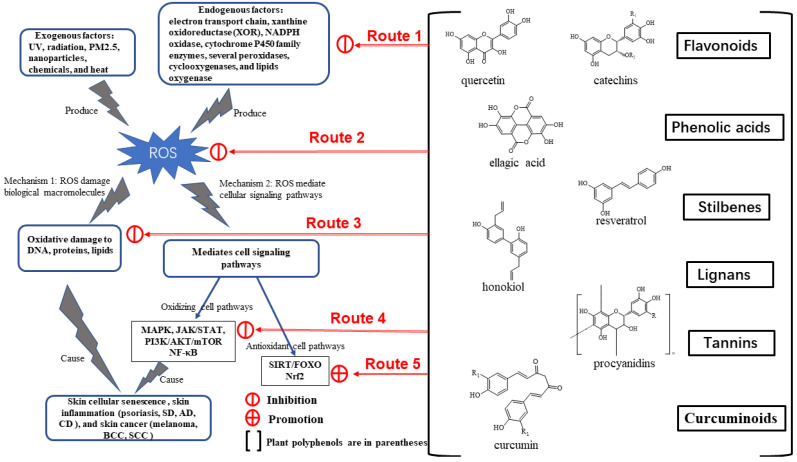
The regulatory mechanism of plant polyphenols.

**Table 1 ijms-24-03755-t001:** Summary of exogenous factors and adverse consequences of ROS production in some experiments.

ROS Stimulators	Experimental Models	Experimental Results	References
UV	UVA (15 K/cm^2^)-induced HaCaT cells, UVA (15 J/cm^2^/every 2 days/14 days)-induced nude mice	UVA toxicity, DNA single-strand breaks, apoptotic DNA fragmentation, dysregulated Bax/Bcl-2 ratio	[25]
	UVB (15 mJ/cm^2^, 24 h)-irradiated human dermal fibroblasts	Photoaging: increased production of MMP-1, decreased Nrf2 protein levels, ERK and JNK phosphorylation	[26]
	Mice exposed to 1700 J/m^2^ UVB radiation four times per day.	Inflammation and immune response	[27]
Radiation	The left thigh skin of mice irradiated with X-ray (40 Gy 180 kV) 5 days a week for 1 month	Increased fibrosis, inflammation, and oxidative stress injury indices, and decreased expression of Nrf2 and its regulatory antioxidant enzymes	[28]
	Human skin cells induced by gamma-ray	Production of p53, p21, oxidative stress markers, and apoptosis and expression of MMP and cytokine genes	[29]
PM2.5	HaCaT cells induced with 100 μg/mL PM2.5 for 24 h	Inflammation: upregulation of the inflammasome NLRP1 and IL-1β expression via ROS/NF-κB	[30]
	HaCaT cells cultured with 50 μg/mL PM2.5 for 7 days	Skin cellular senescence: decreased DNA methyltransferase expression, increased DNA demethylase, decreased histone H3 lysine 27 trimethylation (H3K27Me3)	[31]
	BALB/c mice were treated with PM2.5 solution (0.187, 0.375, and 0.75 mg/kg body weight (b.w.)) or saline through a pipette tip into the nose for fifteen days; The RBL-2H3 cells treated with PM2.5 (0, 25, 50, or 100 mg/mL) for 24 h	Increased cytokine expression in mast cells; MEKK4 and JNK1/2 were activated	[32]
	Human epidermal melanocytes treated with PM2.5 at various concentrations (0, 1, 10, 100, and 250 µg/mL) for 48 h	Inhibited the proliferation of melanocytes and induced their apoptosis	[33]
Nanoparticles	Cationic AuNPs with a diameter of 25 nm induced cellular stress in macrophages	Increased cytokine expression	[34]
	Human skin melanoma (A375) cells induced by cerium oxide nanoparticles	Cytotoxicity, malondialdehyde, and SOD production; decreased glutathione levels; DNA double-strand breaks	[35]
	Human skin cells (A431) induced by nickel carriers (NiNPs)	Lipid peroxidation; genotoxicity; apoptosis; and increased catalase, SOD, and caspase-3 activities	[36]
Chemicals	Hydrogen peroxide-induced injury of human skin fibroblasts	Human skin fibroblasts have decreased viability, cell apoptosis	[37]
	Imiquimod-induced mice	Psoriasis	[38]
	NC/Nga mice induced by 2, 4-dinitrochlorobenzene	Atopic dermatitis	[39]
	Mice induced by 12-O-tetracyl-cutoene-13-acetate	Irritant contact dermatitis	[40]
	Rat model exposed to trivalent arsenic (iAs (3+))	Skin cancer	[41]
Heat	Skin fibroblasts (Hs68) incubated for 30 min at 43 °C	Elevated thiobarbituric acid reactive substances and 8-OH-dG	[42]

**Table 2 ijms-24-03755-t002:** A summary of molecular targets of several polyphenols in preventing skin cellular senescence, inflammation, and cancer.

Polyphenols	Cell or Animal Types	Skin Cellular Senescence, Inflammation, and Cancer	Molecular Targets/Mechanisms	References
Curcumin	HaCaT; Mice	Skin cellular senescence, inflammation, cancer	↓ROS, lipid peroxidation, DNA damage, NADPH oxidase Chelate iron ions	[142,143,144]
	Fibroblasts; Mice; HDFs	Skin cellular senescence	↓ MMPs ↓ MAPK/p38, NF-κB pathways ↑ Collagen I, HO-1 ↑ TGF-β, Nrf2 pathways	[145,146,147]
	Mice	Inflammation (SD)	↓ PI3K/AKT/NF-κB pathway ↑ Nrf2 pathway	[148,149]
	HaCaT	Inflammation (Psoriasis)	↓ MAPK, NF-κB pathways ↓ IL-17, TNF-α, AP-1, IL-22, IL-18 ↓ IFN-γ, IL-6, iNOS	[150,151]
	Mice	Inflammation (AD)	↓ IL-33, IL-4, IL-5, IL-13,IL-31	[152]
	Mice; Hamster	Skin cancer	↓ C-fos, c-jun, c-myc, PCNA, cyclin D1 ↓ Bcl-2 ↑ p53, Box, caspase-3 ↓ MAPK/ERK, JAK/STAT pathways	[114,153,154,155]
	B16	Skin cancer (Melanoma)	↓ NF-κB pathway ↑ PTEN, PDCD4 ↓ Cyclin D1, Ki67	[156]
	A375 and C8161	Skin cancer (Melanoma)	↓ AKT/mTOR pathway ↓ Mcl-1, Bcl-2 ↑ Bax	[149,157]
	Mice	Skin cancer (SCC)	↓ PI3K/AKT/mTOR pathway	[158]
Catechins	Keratinocytes; Rats	Skin cellular senescence, inflammation, cancer	↓ DNA damage, mitochondrial ↓ damage, lipid peroxidation Chelate iron ions	[159,160,161,162,163]
	HaCaT; Mice; HDF	Skin cellular senescence	↓ ROS, MMP-1, MMP-2, MMP-9, AP-1 ↓ IL-1β, IL-6 ↓ MAPK, NF-κB pathways ↓ Lipid peroxidation	[164,165,166,167]
	Mice	Inflammation (CD)	↓ TNF-α, IL-1β, IL-4 ↓ ROS	[168]
	Mice; HaCaT	Inflammation (Psoriasis)	↓ IL-17A, IL-17F, IL-22, IL-23 ↑ SOD, CAT ↓ JAK/STAT pathway ↓ MDA, DNA damage	[169,170,171]
Catechins	Mice	Skin cancer (SCC)	↑ GSH, SOD, CAT, GST, GR, GPx ↑ Nrf2 pathway ↓ COX-2, iNOS, TNF-α, IL-1β, IL-6 ↓ NF-κB pathway ↓ MDA	[172]
	A375 cells	Skin cancer (Melanoma)	↑ PI3K/AKT/mTOR pathway ↓ Bcl-2	[173]
Resveratrol	HaCaT; Mice; Fibroblasts; Keratinocytes	Skin cellular senescence	↑ SOD, GSH-Px, collagen I ↑ SIRT1/FOXO pathway ↓ COX-2, MMPs ↓ JNK MAPK pathway	[174,175,176,177,178]
	Human keratinocytes; Mice	Inflammation (AD)	↓ MAPK pathway ↓ IL-31, IL-8 ↓ ROS, DNA damage ↓ lipid peroxidation	[179,180]
	Mice	Inflammation (Psoriasis)	↓ NF-κB pathway ↓ IL-17A, IL-19	[181]
	Mice	Skin cancer	↑ Bax ↓ Bcl-2	[182]
	Mice	Skin cancer	↑ p53 ↓ MAPK pathway	[183]
	A431	Skin cancer	↓ JAK/STAT pathway	[184]
Quercetin	HST; JB6 P+; HDF; HaCaT; Mice	Skin cellular senescence	↓ ROS, MMP-1, COX-2, AP-1, XOR ↓ NF-κB, MAPK pathways ↑ SIRT1/FOXO pathway	[185,186,187,188,189,190]
	Human keratinoytes	Inflammation	↓ JAK/STAT pathway ↓ IL-1β, IL-18	[191]
	Mice; HaCaT	Inflammation (AD)	↓ NF-κB, MAPK pathways ↓ COX2, TNF-α, IL-1, IL-2R, IL-1β ↓ IL-6, IL-8 ↑ Nrf2 pathway ↑ SOD, CAT, GPx	[192,193]
	Mice	Inflammation (Psoriasis)	↓ TNF-α, IL-6, IL-17 ↑ GSH, CAT, SOD	[194]
	Human melanoma cells	Skin cancer (Melanoma)	↑ Bax ↓ Bcl-2 ↑ Nrf2 pathway ↓ DNA damage	[195,196]
Ellagic acid	Mice; Fibroblasts; Human skin; HaCaT; Macrophages	Skin cellular senescence	↓ IL-1β, IL-6, MMPs, ROS ↑ TGF-β pathway ↓ NADPH oxidase, DNA damage ↑ HO-1, SOD	[197,198,199,200,201]
Ellagic acid	HaCaT; Mice	Inflammation (AD)	↓ MAPK, JAK/STAT pathways	[202,203]
	Mice	Inflammation (CD)	↓ IL-1β, IL-4 ↓ DNA damage, ROS	[204]
	A375; B16	Skin cancer (Melanoma)	↑ Apoptosis ↓ PI3K/AKT/mTOR, NF-κB pathways	[205,206,207]
Proanthocyanidins	HFF-1	Skin cellular senescence	↑ SOD	[208]
	Th17 and Treg cells	Inflammation (Psoriasis)	↓ IL-17, IL-22, TNF-γ, IFN-α, VEGF	[209]
	Mice	Inflammation (CD)	↓ IL-2, IFN-γ, IL-17	[210]
	Mice	Skin cancer (Photocarcinogenesis)	↓ MAPK, NF-κB pathways	[211]
	A375 and Hs294t	Skin cancer (Melanoma)	↓ ERK, MAPK, NF-κB pathways ↓ COX-2	[212]
	Keratinocytic	Skin cancer (SCC)	↑ Autophagy	[213]
Honokiol	HaCaT; HFF-1; Keratinocyte; Mice	Skin cellular senescence	↓ ROS, MMP-1 ↓ NF-κB, MAPK pathways	[214,215,216,217]
	Mice	Inflammation (AD)	↓ IL-4, IL-13, IL-17, TNF-γ	[218]
	HaCat	Inflammation	↓ IL-1, IL-8	[215]
	A431	Skin cancer (SCC)	↓ Cyclin D1, cyclin D2, cdk2, cd	[219]

Abbreviations: ↑: increase; ↓: decrease.

**Table 3 ijms-24-03755-t003:** Plant polyphenol delivery systems (example of curcumin).

Polyphenol	Delivery Systems	Skin Model	Main Results	Potential Therapeutic Application	Reference
Curcumin	Microemulsion	HaCaT cells; Human skin	Significant curcumin concentrations were found in the dermis and curcumin microemulsion-reduced, UV-induced epidermal cytotoxicity.	To protect the skin from oxidative stress-related diseases	[227]
	Phytovesicles	Mice	The phytovesicles were found to be most effective compared to all other formulations and plain curcumin in providing enhanced antioxidant and antiaging effects.	To enhance the antiaging, antioxidant, and antiwrinkle effects of curcumin	[266]
	Elastic vesicle	Mice	Compared with the marketed formula (VICCOA^®^ turmeric skin cream), curcumin ointment (1.64%) had a higher skin retention rate (51.66%). The developed ointment displayed similar effects as the marketed diclofenac sodium ointment (Omni-g (R)) in suppressing acute inflammation in mice.	To treat skin inflammation	[267]
	Liposome (DLs) nanocarriers	Isolated human skin	Continuously penetrated the skin and enhanced its biological properties.	To provide more effective treatment	[268]
	Carbopol 940 hydrogel	Mice	The anti-inflammatory activity of the gel was better than that of the dexamethasone sodium phosphate cream. The hydrogel promoted collagen enrichment and improved the re-epithelialization of wound epidermis.	To treat skin inflammation and full-thickness wound healing	[269]
	Transparent plastid nanovesicles	Human keratinocytes	It protected human keratinocytes from oxidative stress damage in vitro, counteracted inflammation and injury caused by 12-0-tetracyl-chlorowave, reduced edema formation, and improved the biocompatibility and safety of the components.	To increase curcumin’s biocompatibility and safety, as well as its anti-inflammatory activity	[229]
	Nanoemulsion loaded polymeric hydrogel	Mice	Compared with curcumin and betamethasone-17-valerate gels, nanolatex preparations showed faster and earlier healing in psoriatic mice.	To treat psoriasis	[270]
	Peptide-modified curcumin-loaded liposome (CRC-TD-Lip)	Mice	It exhibited high stability and high curcumin encapsulation efficiency, accelerated the transdermal delivery of curcumin, and enhanced the inhibition of psoriasis.	To treat psoriasis	[271]
	Dendritic micellar polymer	B16 and melanoma cells	It has good bioavailability and bioactivity.	To treat melanoma	[272]

**Table 4 ijms-24-03755-t004:** Human clinical trials of plant polyphenols for oxidative stress-induced skin cellular senescence and inflammation.

Participants/ N. (Years).	Products Containing Plant Polyphenols	Topical or Ingested Products	Dosage/ Duration	Outcomes	Potential Therapeutic Application	Reference
Skin of breast cancer patients with radiation/191 (36–81)	Curcumin gel	Topical	(Unknown)/three times a day for a week	Topical curcumin prophylactic therapy may treat radiation dermatitis and pain.	To treat skin inflammation	[273]
Patients with moderate scalp psoriasis/30 (18–75)	Turmeric supplements	Ingested	(Unknown)/twice a day for nine weeks	Dermatology Life Quality Index (DLQI) questionnaire and PASI (Psoriasis Area and Severity Index) scores were assessed and turmeric tonic was found to significantly reduce erythema, dandruff, and skin lesions.	To treat skin inflammation	[274]
Children with AD/64 (2–12)	Multi-herb formula anti-itch cream with 16% turmeric extract and 0.1% turmeric oil	Topical	16% turmeric/twice a day	The treatment group (anti-itching cream) and the control group (Moisturex) exhibited statistically significant improvements in all parameters (subjective itching severity clinical evaluation and health).	To treat skin inflammation	[275]
People with aging skin caused by UV radiation/39 (20–40)	Pomegranate extract rich in ellagic acid made into round tablets	Ingested	High dose (200 mg/d ellagic acid), low dose (100 mg/d ellagic acid)/once a day for four weeks	The results of the questionnaire showed that the decline rates of skin luminance values in the low-dose group and high-dose group were suppressed by 1.35% and 1.73%, respectively, relative to the baseline. In addition, an improvement trend in some items, such as “facial brightness” and “spots and freckles”, was observed.	To treat skin photoaging	[199]
Women with moderate photoaging/10 (unknown)	Green tea oral supplement and green tea cream	Topical and ingested	10% green tea cream and 300 mg twice-daily green tea oral supplementation for eight weeks.	Participants receiving a combined topical and oral green tea regimen showed histological improvements in elastin content, but no clinically significant changes could be detected.	To treat skin photoaging	[276]

## Data Availability

Not applicable.

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
