# Peer review of "Possible Mechanisms of Oxidative Stress-Induced Skin Cellular Senescence, Inflammation, and Cancer and the Therapeutic Potential of Plant Polyphenols"

_ijms, 2023, doi:10.3390/ijms24043755_

Round 1
Reviewer 1 Report
The manuscript entitled “Possible Mechanisms of Oxidative Stress-Induced Skin Aging, Inflammation, and Cancer and the Therapeutic Potential of Plant Polyphenols” and authored by Liu and colleagues first showed how harmful oxidative stress is to the skin contributing to the progression of skin aging, inflammation, and cancer. They then reviewed how plant polyphenols prevent skin damage through their molecular targets thus attracting great interest to their low toxicity. Authors also discussed the most recently discovered skin drug delivery technology for plant polyphenols and how it turned out to be more durable and effective encouraging clinical research on plant polyphenol-based products. It is essential that general background on health-promoting benefits of natural products is provided before focusing on plant polyphenols. The following studies address such shortcoming: http://dx.doi.org/10.4236/jdm.2011.13006, PMID: 17151319, PMID: 32460808, https://doi.org/10.4236/ajps.2018.96091, https://doi.org/10.1186/s41936-020-00177-9, PMID: 33255507, PMID: 22812448. One major concern that should be addressed: What time range of publication did this review article cover, what keywords did the search for literature include, what were the inclusion criteria, how many studies did the search find and how many were primary research vs review articles, of those, how many were selected for evaluation in this study, and finally what criteria were used for selecting the articles that were reviewed (was it the subject of the study, its novelty or both). Other clear setback is lacking in-depth coverage of relevant patents and clinical trials.
Other comments
· Proofreading is required.
· Figures 1 and 3 need more elaborative legends.
· Abbreviations should be revisited. It might be best to add a section for abbreviations.
· Section should be added to shed some light on relevant patents and clinical trials.
· References should be enriched with more diversified investigations. Results from the following studies could serve this purpose: https://doi.org/10.1186/s41936-022-00321-7, PMID: 36432184, PMID: 35740022.
Reviewer 2 Report
This review discusses in detail the therapeutic potential of selected polyphenolic compounds and outline relevant molecular targets. Polyphenols selected here for study according to their structural classification include curcumin, catechins, resveratrol, quercetin, ellagic acid, and procyanidins. Finally, considering curcumin as an example, the latest delivery of plant polyphenols to the skin was summarized, providing a theoretical foundation for future clinical research and the generation of new pharmaceuticals and cosmetic. Overall, the manuscript is well written, though it requires minor grammatical error checks. Their observations are very interesting, comprehensive and summarize significant findings in this review.
Plant polyphenols are a type of remarkable antioxidants in different biological system, including red blood cells, because the ortho-phenolic hydroxyl group in the phenolic hydroxyl structure is easily oxidized into a quinone structure. This structure has a considerable potential to scavenge free radicals such as ROS. I suggest to add this reference (DOI: 10.3390/ijms23147781).
Aging is the progressive loss of tissue and organ function over time. The free radical theory of aging, later termed as oxidative stress theory of aging, is based on the structural damage-based hypothesis that age-associated functional losses are due to the accumulation of oxidative damage to macromolecules by ROS. The exact mechanism of oxidative stress-induced aging is still not clear, but probably increased ROS levels lead to cellular senescence, a physiological mechanism that stops cellular proliferation in response to damages that occur during replication. Senescent cells acquire an irreversible senescence-associated secretory phenotype involving secretion of soluble factors (interleukins, chemokines, and growth factors), degradative enzymes like matrix metalloproteases, and insoluble proteins/extracellular matrix components. I suggest to improve the introduction focuses, especially the key difference between aging and senescence, as they are different mechanism. In general, aging is the process of deterioration of cells with time, while senescence is a result of aging where the cells stop dividing and reach a state of arrest.
I suggest to introduce this recent reference (DOI: 10.3390/ijms222312641).
Reviewer 3 Report
The review articles entitled “Possible Mechanisms of Oxidative Stress-Induced Skin Aging, Inflammation, and Cancer and the Therapeutic Potential of Plant Polyphenols” is complete and well described. This review article discusses the molecular mechanism of skin ageing, inflammation, and cancer, as well as the therapeutic potential of plant polyphenols. This manuscript is attractive to the readership and should be considered for publication. However, this MS has some minor issues that should be addressed before acceptance.
Minor comments
Fig.1: What Is the Cytochrome P45? It has to be changed to 450.
At the line number 228 you should provide the full form of AD.
Reviewer 4 Report
Accept in the present form!
Round 2
Reviewer 1 Report
none